# Effect of the COVID-19 lockdown on the HIV care continuum in Southwestern Uganda: A time series analysis

Jane Kabami[1,2]*, Asiphas Owaraganise[1], Brian Beesiga[1]*, Jaffer Okiring[1], Elijah Kakande[1], Yea-Hung Chen[3], Florence Mwangwa[1], Cecilia Akatukwasa[1], Joanita Nangendo[2], Winnie Muyindike[4], Fred C. Semitala[1,2,5], Michelle E. Roh[3], Moses R. Kamya[1,2]

1 Infectious Diseases Research Collaboration (IDRC), Kampala, Uganda, 2 Department of Medicine, Makerere University, Kampala, Uganda, 3 Department of Epidemiology and Biostatistics, University of California, San Francisco, San Francisco, California, United States of America, 4 Global Health Collaborative, Mbarara University of Science and Technology, Mbarara, Uganda, 5 Makerere University Joint AIDS Program (MJAP), Kampala, Uganda

* kabamijane@gmail.com (JK); beesigab@gmail.com (BB)

## Abstract

### Introduction

In Uganda, COVID-19 lockdown policies curbed the spread of SARS-CoV-2, but their effect on HIV care is poorly understood.

### Objectives

We examined the effects of COVID-19 lockdown policies on ART initiation, missed visits, and viral suppression in Uganda.

### Methods

We conducted a time series analysis using data from a dynamic cohort of persons with HIV enrolled between March 2017 and September 2021 at HIV clinics in Masaka and Mbarara Regional Referral Hospitals in Southwestern Uganda. Poisson and fractional probit regression were used to predict expected monthly antiretroviral therapy initiations, missed visits, and viral suppression based on pre-lockdown trends. Observed and expected trends were compared across three policy periods: April 2020-September 2021 (overall), April-May 2020 (1st lockdown), and June-August 2021 (2nd lockdown).

### Results

We enrolled 7071 Persons living with HIV (PWH) (n_Masaka = 4150; n_Mbarara = 2921). Average ART duration was 34 and 30 months in Masaka and Mbarara, respectively. During the 18-month post-lockdown period, monthly ART initiations were lower than expected in both Masaka (51 versus 63 visits; a decrease of 12 [95% CI: -2, 31] visits) and Mbarara (42 versus 55 visits; a decrease of 13 [95% CI: 0, 27] visits). Proportion of missed visits was moderately higher than expected post-lockdown in Masaka (10% versus 7%; 4% [95% CI: 1%,

**Funding:** MK and FS under Award Number D43 TW010037. Research reported in this publication was supported by the Fogarty International Center and Office of AIDS Research of the National Institutes of Health http://grants.nih.gov/grants/policy/coi/ The content is solely the responsibility of the authors and does not necessarily represent the official views of the National Institutes of Health.

**Competing interests:** The authors have declared that no competing interests exist.

7%] absolute increase), but not in Mbarara (13% versus 13%; 0% [95% CI: -4%, 6%] absolute decrease). Viral suppression rates were moderate-to-high in Masaka (64.7%) and Mbarara (92.5%) pre-lockdown and remained steady throughout the post-lockdown period.

## Conclusion

The COVID-19 lockdown in Uganda was associated with reductions in ART initiation, with minimal effects on retention and viral suppression, indicating a resilient HIV care system.

## Introduction

On March 11, 2020, the World Health Organization (WHO) declared COVID-19 a global pandemic [1, 2]. In response, the Ugandan government undertook several public health response strategies to prevent the spread of COVID-19 [3, 4]. From March 30, 2020, the Government of Uganda instated a full lockdown lasting ten weeks with restrictions on mass gatherings, closure of schools, closure of all border points except for cargo trucks, a ban on public transport, and a nationwide curfew coinciding with the first wave of COVID cases reported in the country [5]. Heavy restrictions remained in place until June 4, 2020, when some of the restrictions were relaxed. Importantly, public transport resumed at half capacity in non-border districts. In response to the second wave of the COVID pandemic [6], the government announced a second lockdown on June 6, 2021, which included greater restrictions on public transport, school closures, a ban on public gatherings, and a nationwide curfew. This lockdown period lasted 12 weeks until restrictions were relaxed on September 2, 2021.

While the lockdown policies were effective in limiting the spread of COVID-19, they may have had unintended consequences on HIV service delivery. In Uganda, where persons living with HIV (PWH) make up approximately 1.3 million of the total population, regular and timely attendance to HIV clinics is necessary for the provision of life-saving antiretroviral therapy (ART) [7, 8]. Disruptions in care can have major ramifications, undoing the tremendous progress attained in improving the HIV care continuum in Uganda [9]. While several studies evaluated this phenomenon, findings from these studies were mixed, with some reporting a negative effect on HIV care outcomes [10, 11] and others reporting little or no effect [12, 13]. Furthermore, these studies were conducted in countries other than Uganda, most prominently in South Africa, where the lockdown policies were the strictest and the prevalence of HIV is among the highest in the world. In Uganda, a few studies have evaluated the effect of the lockdown on HIV care showed an increase in missed visits by 15–50% and no effect on viral suppression [7, 11], however, none of these studies conducted a rigorous longitudinal evaluation of the impact of the COVID-19 lockdown policies on multiple outcomes on the HIV care continuum. These data are needed to inform strategies to ensure continuity of care during future pandemics.

Our study aimed to determine the effect of the COVID-19 response on ART initiation, missed visits, and viral suppression among PWH receiving care at two high-volume Regional Referral Hospitals in Southwestern Uganda.

## Materials and methods

### Study setting

The study was conducted in the Southwestern Region of Uganda, where the adult HIV prevalence is 6.3% [14]. In this region, there are two large HIV clinics embedded within Masaka and

Mbarara Regional Referral Hospitals. Each hospital enrols more than 1000 HIV patients into care per year. The two hospitals participate in the East African International Epidemiology Databases to Evaluate AIDS (IeDEA-EA), a multi-centre consortium of HIV care and treatment centres whose overall goal is to improve HIV data quality globally [15].

Each PWH enrolling into HIV care is reviewed by a clinician who reviews their medical history to initiate the most appropriate ART regimen and develop a management plan according to the national HIV prevention and treatment guidelines [16]. PWH are then scheduled for follow-up visits every three to six months. For stable adult clients, viral load measurements are performed six months after ART initiation and then annually thereafter, while for unstable patients, children, adolescents, and pregnant women, viral load measurements are conducted more frequently per the treatment guidelines [16].

## Data source and data extraction

We extracted individual-level data from electronic medical records (EMR) of PWH attending HIV clinics at Masaka and Mbarara Regional Referral Hospitals between March 2017 and September 2021, 36 months before and 18 months after the first lockdown policy in March 2020. At each of the study sites, patient data are routinely collected in standardized Ministry of Health forms and checked by data managers for completeness and accuracy before being entered in EMR.

To be included in the current study, PWH were required to have a confirmed HIV diagnosis, have at least one viral load measurement during the study period, and must have initiated ART during the study period (March 2017-September 2021). Those who died before the start of the pandemic were excluded from the analysis.

We extracted data on age, sex, marital status, number of clinic visits, ART start date, appointment/scheduled visit date, actual visit date, and viral load result.

## Measurement of the policy period

The main objectives of this study were to evaluate the effect of the COVID-19 lockdown policies on ART initiation, missed visits, and viral suppression. Given that two lockdown policies were implemented throughout the study period, three policy periods were evaluated: (1) the overall effect defined as the 18-month period after the first lockdown (April 2020-September 2021); (2) the two-month period during the first lockdown policy (April 2020-May 2020); and (3) the three-month period during the second lockdown policy (June-August 2021).

## Measurement of outcomes

The outcomes evaluated were number of ART initiations per month, proportion of missed visits per month, and proportion virally suppressed per month. Outcomes were assessed at the individual level and then aggregated by month to perform analyses. A visit was considered missed if PWH did not come to the clinic after 14 days from their scheduled visit date. Viral suppression was defined as a viral load test result that indicated <1000 copies/ml per the national HIV guidelines.

## Statistical analysis plan

The study used time series analyses to estimate the effect of the COVID-19 lockdown policies on key HIV service delivery indicators. Prior to conducting our time series analyses, we found substantial differences in pre-lockdown outcomes (e.g., number of ART initiations, viral suppression rates) between study sites. Thus, analyses were conducted separately for Mbarara and

Masaka Referral Hospitals to explore whether the effects of the lockdown varied between study sites. To estimate the impact of the COVID-19 lockdown policy, we assumed that pre-lockdown trends would have resumed had the lockdown policies never been instated [17, 18]. To estimate the pre-lockdown trend, the following model was used: $Y_t = \beta_0 + \beta_1 T_t + \beta_2 C_t$, where for time points $t$, $Y$ represents the outcome; $T$ is a linear term indicating the number of months since the start of the study period (models the linear pre-lockdown trend) and $C$ is an indicator variable for calendar month (e.g., January, February, March etc.) to account for seasonality. Poisson regression was used to model counts, and fractional probit regression was used to model proportions. For Poisson models, a stable target population was assumed and thus, no offset term was specified. Sensitivity analyses using an autoregressive integrated moving average (ARIMA) model were conducted (S1 Appendix).

Models were then used to predict the expected monthly outcomes in the absence of the COVID-19 lockdown policies for each month during the post-lockdown period. To estimate the impact of the lockdowns, relative and absolute differences between observed and expected values were compared for each post-policy month and averaged across each post-policy period (e.g., overall, first and second lockdown periods). 95% confidence intervals around effect estimates were obtained using a percentile-bootstrapping procedure of 1,000 simulations. Analyses were conducted using Stata 16.1 (College Station, Texas, USA) and R version 4.0.1 (Vienna, Austria).

### Ethical approval

This study was approved by the Uganda National Council of Science and Technology and the School of Medicine Research and Ethics Committee, Makerere University. We obtained a waiver of consent because we utilized secondary data.

## Results

### Characteristics of the study population

Over the study period, we included data of 7071 PWH, 4150 (58.7%) from Masaka Referral Hospital, and 2921 (41.3%) from Mbarara Referral Hospital. Among PWH enrolled at Masaka Referral Hospital, nearly two-thirds were female (65.8%). Median age was 30 years (Interquartile range (IQR): 23–37]), and the average duration on ART was 34 months. Among those enrolled at Mbarara Referral Hospital, median age was 30 years (IQR: 25–38), 62.2% were female, and the average duration on ART was 30 months. In both sites, nearly half of the patients were aged 18 to 30 years (53.6% in Masaka and 48.2% in Mbarara) (Table 1).

### Trends in ART initiation, missed visits, and viral suppression prior to lockdown

The assessment of monthly trends in ART initiation, missed visits, and viral suppression prior to the lockdown is shown in Fig 1. Prior to the pandemic and associated lockdown policies, monthly trends in ART initiations were declining steadily in both Masaka and Mbarara Referral Hospitals. Average monthly number of ART initiations were higher in Masaka than in Mbarara Referral Hospital (87.4 and 57.4, respectively) (Table 2). The monthly proportions of missed visits were generally low at both Masaka and Mbarara Referral Hospitals and accounted for less than 10% of the total visits at both hospitals. Monthly proportions of virally suppressed patients were lower in Masaka than in Mbarara Referral Hospital (64.7% versus 92.5%, respectively), and trends remained fairly steady during the pre-lockdown period.

**Table 1. Description of the study population.**

| Characteristic | Site | |
|---|---|---|
| | **Masaka** | **Mbarara** |
| Total PWH | 4150 | 2881 |
| Female, n (%) | 2729 (65.8) | 1816 (63.0) |
| Median age in years (IQR) | 30 [23–37] | 30 [25–38] |
| Age categories, n (%)[1] | | |
| <17 | 234 (5.6) | 10 (0.4) |
| 18–30 | 2223 (53.6) | 1406 (55.5) |
| 31–40 | 1096 (26.4) | 681 (26.9) |
| 41–50 | 436 (10.5) | 282 (11.1) |
| ≥51 | 161 (3.9) | 153 (6.0) |
| Marital status, n (%)[2] | | |
| Single/Never married | 563 (15.8) | 578 (22.9) |
| Married | 1867 (52.5) | 1362 (54.0) |
| Separated/divorced | 1127 (31.7) | 581 (23.0) |
| Number of ART clinic visits, mean (SD) | | |
| Pre-lockdown (March 2017-March 2020) | 12.5 (4.8) | 13.4 (5.7) |
| Post-lockdown (April 2020-September 2021) | 12.6 (4.1) | 10.8 (5.5) |
| Duration on ART in months, mean (SD)[3] | 34.2 (16.0) | 29.7 (15.3) |
| Total missed visits by all PWH | 2284 | 4195 |

Abbreviations: ART = antiretroviral therapy; IQR = interquartile range

[1] Missing age for 389 patients attending Mbarara Referral Hospital.

[2] Missing marital status for 593 patients attending Masaka Referral Hospital and for 400 patients attending Mbarara Referral Hospital

[3] Calculated over the entire study period (March 2017 to September 2021)

## Trends in ART initiation, missed visits, and viral suppression post-lockdown

**Masaka Referral Hospital.** Assuming pre-lockdown trends would have continued throughout the post-lockdown period, there were 12 fewer ART initiations per month [95% CI: 2, 31] during the 18-month period after the first lockdown was initiated (Table 2, Fig 2A). Larger reductions were observed when lockdowns were in place (14 [95% CI: 2, 31] fewer ART initiations per month during the first lockdown period and 27 [95% CI: 2, 59] fewer ART initiations per month during the second lockdown).

While missed visits were low across the entire study period, a greater proportion of scheduled visits were missed per month during the 18-month post-lockdown period (10.4% versus 6.5%; a 3.9% [95% CI: 0.8, 6.6] absolute increase) (Table 2, Fig 2A). The impact on missed visits were primarily during the periods when lockdowns were imposed. Effects were larger during first lockdown compared to the second lockdown; a nearly 3.43-fold increase in missed visits during the first lockdown compared to a 1.5-fold increase in the second lockdown) (Table 2, Fig 2A).

Monthly rates of viral suppression at Masaka Referral Hospital ranged from 56.0% to 77.6% across the entire 18-month post-lockdown period. Trends in viral suppression were similar to what was expected prior to the pandemic (mean absolute increase of 1.8% [95% CI: -9.0%, 12.3%]) (Table 2), with minimal differences observed during the periods when the first and second lockdowns were imposed.

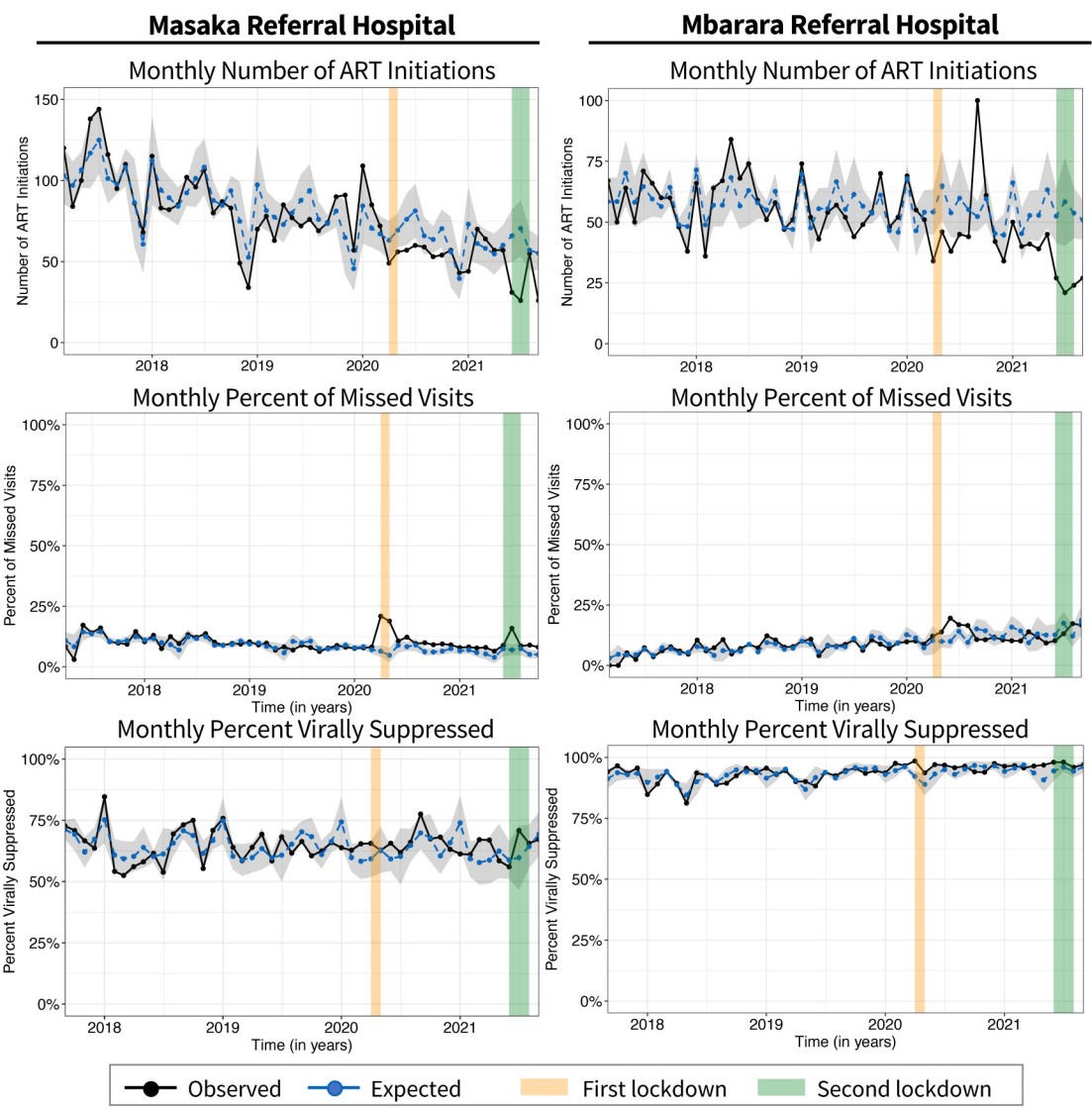

**Fig 1. Observed and expected monthly trends in HIV service delivery indicators before and after COVID-19 lockdown policies.** Observed trends are shown in black. Expected trends which were estimated based on pre-COVID-19 lockdown trends are shown in blue. Grey-shaded areas represent 95% confidence intervals. Orange and green shaded areas represent the period when the first and second lockdown policies were imposed.

**Mbarara Referral Hospital.** In the 18 months following the first lockdown, ART initiations were 23% [95% CI: 0%, 39%] lower than expected at Mbarara Referral Hospital (42 versus 55 initiations). Similar to Masaka Referral Hospital, larger reductions in ART initiations were seen during the periods when the first and second lockdowns were imposed (Table 2, Fig 2B).

Across the 18-month post-lockdown period, the average observed proportions of missed visits in Mbarara were similar to what was expected based on pre-lockdown trends (13.0% vs 12.9%; a 0.2% [95% CI -4.0, 5.9] absolute decrease). The average proportion of missed visits was higher than expected when the first lockdown period was imposed (13.0% vs. 10.1%; a 2.9% [95% CI: -3.4, 8.9] absolute increase), but this finding did not reach statistical significance.

**Table 2. Comparison of observed and expected outcomes of HIV service delivery indicators before and after COVID-19 lockdown policies in Masaka and Mbarara Referral Hospitals.** Expected outcomes were estimated based on the pre-COVID-19 lockdown trend modeled using Poisson regression (for counts) or fractional probit regression (for proportions).

| | Observed | Expected* | Absolute difference [95% CI] | Relative difference [95% CI] |
|---|---|---|---|---|
| **Masaka Referral Hospital** | | | | |
| *Monthly ART initiations, n (SD)* | | | | |
| Pre-lockdown | 87.4 (22.5) | 87.4 (17.5) | 0 [0, 0] | 1.00 [0.99, 1.00] |
| Overall | 51.0 (12.5) | 63.4 (9.6) | -12.4 [-31.0, 2.1] | 0.82 [0.63, 1.07] |
| 1st lockdown | 52.5 (4.9) | 66.3 (4.4) | -13.8 [-31.1, -1.6] | 0.79 [0.63, 0.97] |
| 2nd lockdown | 37.3 (15.5) | 64.5 (6.8) | -27.1 [-59.4, -1.8] | 0.60 [0.33, 0.98] |
| *Missed visits per month, % (SD)* | | | | |
| Pre-lockdown | 9.9% (2.9) | 9.9% (2.1) | 0% [0, 0] | 1.00 [1.00, 1.01] |
| Overall | 10.4% (3.9) | 6.5% (1.4) | 3.9% [0.8, 6.6] | 1.66 [1.14, 3.21] |
| 1st lockdown | 19.9% (1.4) | 5.6% (1.2) | 14.0% [11.4, 16.7] | 3.42 [2.27, 6.78] |
| 2nd lockdown | 11.1% (4.2) | 7.4% (0.3) | 3.7% [-2.4, 8.4] | 1.54 [0.81, 2.62] |
| *Virally suppressed per month, % (SD)* | | | | |
| Pre-lockdown | 64.7% (7.3) | 64.7% (5.1) | 0% [0, 0] | 1.00 [1.00, 1.00] |
| Overall | 65.1% (4.8) | 64.1% (4.7) | 1.1% [-9.0, 12.3] | 1.02 [0.89, 1.26] |
| 1st lockdown | 64.2% (2.0) | 61.1% (2.6) | 3.1% [-12.4, 16.2] | 1.05 [0.85, 1.33] |
| 2nd lockdown | 64.2% (7.5) | 60.9% (3.0) | 3.2% [-13.5, 20.3] | 1.05 [0.83, 1.45] |
| **Mbarara Referral Hospital** | | | | |
| *Monthly ART initiations, n (SD)* | | | | |
| Pre-lockdown | 57.4 (10.8) | 57.4 (7.4) | 0% [0, 0] | 1.00 [1.00, 1.00] |
| Overall | 42.1 (17.5) | 54.7 (6.4) | -12.6 [-26.5, -0.1] | 0.77 [0.61, 1.00] |
| 1st lockdown | 40.0 (8.5) | 59.5 (7.6) | -19.5 [-36.9, -8.3] | 0.67 [0.51, 0.84] |
| 2nd lockdown | 24.0 (3.0) | 54.8 (3.1) | -30.8 [-52.5, -17.8] | 0.44 [0.31, 0.59] |
| *Missed visits per month, % (SD)* | | | | |
| Pre-lockdown | 7.4% (2.9) | 7.4% (2.4) | 0% [0, 0] | 0.99 [0.92, 1.00] |
| Overall | 13.0% (3.1) | 12.9% (2.0) | -0.2% [-5.9, 4.0] | 1.07 [0.69, 1.40] |
| 1st lockdown | 13.0% (1.2) | 10.1% (0.3) | 2.9% [-3.4, 8.9] | 1.29 [0.68, 1.42] |
| 2nd lockdown | 13.5% (3.5) | 14.1% (2.9) | -0.5% [-9.8, 6.3] | 1.00 [0.57, 1.73] |
| *Virally suppressed per month, % (SD)* | | | | |
| Pre-lockdown | 92.5% (3.6) | 92.5% (2.7) | 0% [0, 0] | 1.00 [0.99, 1.00] |
| Overall | 96.4% (1.4) | 94.4% (2.2) | 2.0% [-0.6, 6.6] | 1.02 [0.99, 1.07] |
| 1st lockdown | 96.1% (3.5) | 90.6% (2.3) | 5.5% [0.4, 10.9] | 1.06 [1.00, 1.13] |
| 2nd lockdown | 97.3% (1.2) | 95.0% (0.9) | 2.4% [-0.5, 7.7] | 1.03 [0.99, 1.09] |

* Derived from the linear regression model extrapolating pre-lockdown trends onto the post-lockdown period.

Monthly rates of viral suppression at Mbarara Referral Hospital ranged from 93.7% to 98.6% across the entire 18-month post-lockdown period. Trends in viral suppression during the 18-month post-lockdown period were similar to pre-pandemic levels and did not substantively differ when the two lockdown policies were imposed.

## Discussion

Overall, our study revealed that the COVID-19 lockdown policies implemented in Uganda between April 2020 to September 2021 were associated with lower monthly ART initiations at both Masaka and Mbarara Referral Hospitals, an increase in missed visits at Masaka Referral

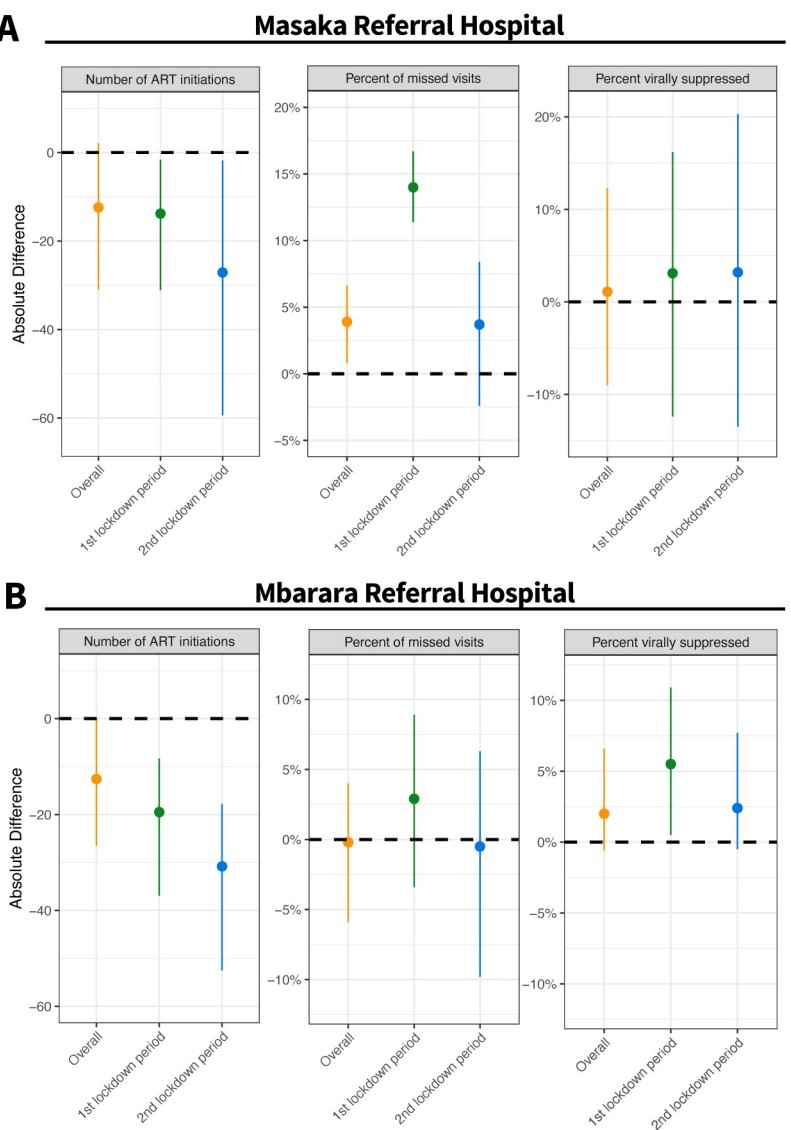

**Fig 2. Predicted impact of COVID-19 lockdown policies on indicators of HIV service delivery at Masaka (Panel A) and Mbarara Referral Hospitals (Panel B).** In the figure, points indicate absolute differences, and bars indicate 95% confidence intervals.

Hospital (but not Mbarara Referral Hospital), and no change in viral suppression rates in both HIV clinics.

Analyses of monthly ART initiations revealed that greater reductions were observed during the period when the first and second lockdown policies were imposed. The decline in monthly ART initiations at both study clinics when the lockdown policies were imposed could be explained by strict restrictions on transport and movement as well as shelter-in-place orders that impaired access to HIV treatment facilities [19]. The limited access to health services due to travel restrictions may have resulted in a reduction in HIV testing [20], an important step for entry into HIV care. Additionally, the Ministry of Health suspended all community HIV testing services during the period of the pandemic [21]. The overall effect observed was a 41%

reduction in HIV testing in 2020 as a result of the COVID-19 pandemic [22]. Our findings are consistent with prior studies describing reductions in HIV testing and ART initiation as a result of the COVID-19 pandemic. In Kenya, Muhula and colleagues reported a 48% drop in ART initiation during the first wave of the pandemic [23]. In South Africa, an interrupted time series analysis assessing the impact of the COVID-19 pandemic on HIV care showed a 46.2% decrease in ART initiation in the first week following a national lockdown, recovering up to 24.7% of pre-lockdown levels four months after the lockdown [13].

We found that an increase in the proportion of missed visits during the first and second lockdowns at Masaka Referral Hospital, although effects were lower during second lockdown. However, this effect was not observed at Mbarara Referral Hospital, suggesting lockdown policies may impact HIV delivery systems differently. After the introduction of the lockdown, the Ministry of Health issued guidance on the continuity of HIV care and essential services during the pandemic [21]. Innovations such as multi-month dispensing where stable clients were given up to six months of drug supply, scaling up of differentiated service-delivery models such as Community Drug Distribution Points and Community Client led ART delivery, and home delivery of ART ensured that PWH continued to receive care during the pandemic. Additionally, the Ministry of Health introduced a "visitor program" that allowed PWH to get ART refills at the clinics closer to their home. These clinics would coordinate with the patient's primary clinics to update their data in the system [21]. Indeed, Schwartz et al. reported that 49%–66% of those who missed appointments sought care at other health facilities nearby their homes, and 92%-100% of all who sought care accessed medicines [7]. Such efforts may have ensured that patients from our study sites received continuous ART delivery and were subsequently retained in care, particularly in the subsequent lockdown. Our findings from Masaka Referral Hospital with previous studies measuring the effect of the lockdown policies on clinic visit attendance, a proxy for missed visits, reported an increase in missed visits ranging from 15%-50% [10, 11]. Our findings from Mbarara Referral Hospital is consistent with findings from South Africa where studies were conducted in multiple sites, there was a marginal increase [13] or no change in clinic visit attendance [12].

The lockdown policy did not significantly affect viral suppression rates in our study partly because changes in viral suppression take a longer time to be detected but also because of the continued ART supply and retention of PWH during the lockdown period. Evidence from modeling studies suggested that there might be adverse effects of the COVID-19 response on the HIV care continuum in the long term [24]. Thus, future studies using a longer observational period of post-lockdown are needed.

Our study's strengths include a robust time series analysis that allowed us to assess the temporal trends in HIV care before and after the lockdown, a large sample size of 7,071 PWH at two high-volume HIV care delivery centers, and a long period of follow-up to assess the impact of the pandemic over two lockdown periods.

There were limitations as well. First, we cannot rule out unmeasured confounding of our effect estimates. This would be especially true if other interventions other than the lockdown were co-occurring that could have also impacted trends in HIV care service delivery. However, we are not aware of any such interventions. Second, the nuanced details of each lockdown (e.g., the partial lifting of some restrictions) and the actual timing of when communities responded made the designation of the true lockdown periods challenging. Third, impact estimates are highly dependent on the statistical model used to establish expected trend. We conducted sensitivity analyses using an ARIMA model which showed similar results for most outcomes, except for two outcomes; ARIMA models suggest COVID-19 was associated with a greater reduction in ART initiations in Masaka Referral Hospital and more missed visits in Mbarara Referral Hospital compared to our current approach. While we believe that our

estimates of the expected trend are more plausible for reasons outlined in S1 Appendix, the estimates produced for these outcomes should be interpreted with caution (as the true counterfactual trend is unverifiable). Lastly, we focused on collecting high-quality data from large HIV clinics, and thus our estimates may not be generalizable to lower-level health facilities. However, we believe that practices did not differ significantly since the same HIV-implementing partners in that region support the lower facilities under each tertiary hospital. Izudi and colleagues described similar adaptions at lower-level health facilities in Uganda [25].

## Conclusion

Our results highlight that, in Southwestern Uganda, the COVID-19 lockdown significantly affected ART initiation (entry into HIV care) but had little to no impact on viral suppression along the HIV treatment cascade. The impact on missed visits varied between hospitals and were fairly short-lived after the lockdown policies were lifted. Overall, our findings demonstrate the resilience of HIV care service delivery for PWH already engaged in care, which may have been influenced by innovative interventions that were supported by the Ministry of Health to reach PWH during lockdown periods. However, HIV care programs need to develop innovative practices and strategies for sustaining care entry and ensure PWL retain care during future epidemics.

## Supporting information

**S1 Appendix. Alternative model specification using autoregressive integrated moving average (ARIMA) model.**
(DOCX)

**S1 File.**
(ZIP)

## Acknowledgments

We acknowledge the management at Mbarara and Masaka Regional Referral hospitals for enabling us to access patient data and the data managers at these hospitals for their help in the data extraction.

## Author Contributions

**Conceptualization:** Jane Kabami, Asiphas Owaraganise, Elijah Kakande, Florence Mwangwa, Cecilia Akatukwasa, Joanita Nangendo, Fred C. Semitala, Moses R. Kamya.

**Data curation:** Yea-Hung Chen, Michelle E. Roh.

**Formal analysis:** Jane Kabami, Asiphas Owaraganise, Brian Beesiga, Jaffer Okiring, Elijah Kakande, Yea-Hung Chen, Florence Mwangwa, Joanita Nangendo, Michelle E. Roh.

**Funding acquisition:** Joanita Nangendo, Fred C. Semitala, Moses R. Kamya.

**Investigation:** Asiphas Owaraganise, Moses R. Kamya.

**Methodology:** Asiphas Owaraganise, Brian Beesiga, Jaffer Okiring, Elijah Kakande, Florence Mwangwa, Joanita Nangendo, Winnie Muyindike, Fred C. Semitala, Michelle E. Roh, Moses R. Kamya.

**Supervision:** Winnie Muyindike.

**Validation:** Yea-Hung Chen, Michelle E. Roh.

**Writing – original draft:** Jane Kabami, Brian Beesiga, Elijah Kakande, Cecilia Akatukwasa, Fred C. Semitala, Michelle E. Roh.

**Writing – review & editing:** Jane Kabami, Asiphas Owaraganise, Brian Beesiga, Jaffer Okiring, Elijah Kakande, Yea-Hung Chen, Florence Mwangwa, Cecilia Akatukwasa, Joanita Nangendo, Winnie Muyindike, Fred C. Semitala, Michelle E. Roh, Moses R. Kamya.

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
