## [Decision Letter · Decision Letter 0]

3 Feb 2023

PONE-D-22-34356Full Title: Effect of the COVID-19 lockdown on the HIV care continuum in Southwestern Uganda: An interrupted     time series analysisPLOS ONE

Dear Dr. Kabami,

Thank you for submitting your manuscript to PLOS ONE. After careful consideration, we feel that it has merit but does not fully meet PLOS ONE’s publication criteria as it currently stands. Therefore, we invite you to submit a revised version of the manuscript that addresses the points raised during the review process.

We look forward to receiving your revised manuscript.

Kind regards,

Jake Michael Pry, PhD, MPH

Academic Editor

PLOS ONE

Journal Requirements:

3.In your Data Availability statement, you have not specified where the minimal data set underlying the results described in your manuscript can be found. PLOS defines a study's minimal data set as the underlying data used to reach the conclusions drawn in the manuscript and any additional data required to replicate the reported study findings in their entirety. All PLOS journals require that the minimal data set be made fully available. For more information about our data policy, please see http://journals.plos.org/plosone/s/data-availability.

Additional Editor Comments:

We received very different reviews of this work from the reviewers. Reviewer 1 has outlined important areas to revise while Reviewer 2 confirms the value of this study with a minor request for clarification. We would like to invite you to submit a revision given feedback from reviewers.

Thank you for submitting to PLOS One and we look forward to receiving your revision.

Reviewers' comments:

Reviewer's Responses to Questions

**Comments to the Author**

1. Is the manuscript technically sound, and do the data support the conclusions?

Reviewer #1: Partly

Reviewer #2: Yes

2. Has the statistical analysis been performed appropriately and rigorously? 

Reviewer #1: Yes

Reviewer #2: Yes

3. Have the authors made all data underlying the findings in their manuscript fully available?

Reviewer #1: No

Reviewer #2: Yes

4. Is the manuscript presented in an intelligible fashion and written in standard English?

Reviewer #1: Yes

Reviewer #2: Yes

5. Review Comments to the Author

Reviewer #1: Overall, this is a clearly-written and straightforward manuscript that presents relevant data on the impact of the COVID-19 pandemic lockdowns on engagement in HIV care. The findings are timely and relevant both to the literature examining the impact of lockdowns as well as how to best deliver care in the early differentiated service delivery. I have several specific critiques and a few minor points:

1. The authors’ choice to examine two separate lockdowns (April-May 2020 and June-August 2021), in addition to the “overall” period of April 2020 - September 2021, is confusing. The scientific rationale for examining the “overall” period in addition to the actual lockdowns is unclear. Additionally, it is not obvious from reading the manuscript whether they actually conducted three separate interrupted time series analyses (comparing baseline to each of these periods), or somehow incorporated all of the time periods into a single analysis.

2. The segmented regression model correctly includes coefficients for linear time and time since the start of the first lockdown, but I believe that it should also include a non-linear coefficient indicating whether the period is prior to or during lockdown (See Jo Y, et al JIAS 2021, or Ross J, et al JIAS 2019).

3. What was the rationale for stratifying the analysis by site (Mbarara and Masaka)? Table 1 suggests that the populations were quite similar, and presumably they experienced lockdowns at the same time. So why not combine the data?

4. In the Discussion, the authors do a nice job describing steps that the Ugandan HIV program took to counter potential effects of the lockdown on missed visits (e.g. Community Distribution Points, multi-month dispensing), which may have contributed to relatively few missed visits. However, I wonder if the authors also looked at the metric of number of *scheduled* visits in addition to missed visits. It would be useful to know whether fewer visits occurred overall during the lockdowns - if this was the case, but clinical outcomes were not impacted, this would help make a case for lower-barrier DSD models.

5. Similarly, it is promising to see high levels of viral suppression, but it is possible that a substantially fewer number of people had a viral load *measured* during the lockdown periods. It would be helpful for the authors to present data on the number of viral loads in addition to the proportion suppressed.

Minor comments:

6. Intro, paragraph 2, line 73: Can the authors briefly note the findings of the cited studies (references 7 and 11)?

7. The authors should not refer to patients as “participants” in the manuscript, as they utilized routine clinical data with a waiver of informed consent.

8. In the Methods the authors should describe/define all variables that were extracted from the EMR (e.g. sex, age, marital status) in addition to outcomes.

9. In Table 1, was the mean number of ART clinic visits extracted over the entirety of the study, or some sort of baseline measurement? If this was over the entire study period, as noted above, would be helpful to see the mean number pre-lockdown and then during the lockdown. Similarly, was duration on ART a baseline value?

10. In Table 2, it would be helpful to have a footnote for “Expected” indicating that this value was derived from the linear regression model

11. The authors state that all data is available but this does not seem to be the case.

Reviewer #2: Thank you for the opportunity to review this well written manuscript. The study design and analysis approach is appropriate. The ethical considerations were made. The authors have endeavoured to provide contextual information on how the lockdowns were implemented, the study setting and the type of HIV program innovations which are all critical in understanding the findings of this study. The study strengths and limitations are described.

The only question that remains is how other facilities , which are not part of the IeDEA collaboration, would perform with regard to implementing innovations for retention for people living with HIV/AIDS.

6. PLOS authors have the option to publish the peer review history of their article (what does this mean?). If published, this will include your full peer review and any attached files.

Reviewer #1: No

Reviewer #2: **Yes: **Pascalina Chanda-Kapata

---

## [Author Response · Author response to Decision Letter 0]

23 Mar 2023

Reviewer #1: Overall, this is a clearly-written and straightforward manuscript that presents relevant data on the impact of the COVID-19 pandemic lockdowns on engagement in HIV care. The findings are timely and relevant both to the literature examining the impact of lockdowns as well as how to best deliver care in the early differentiated service delivery. I have several specific critiques and a few minor points:

1. The authors’ choice to examine two separate lockdowns (April-May 2020 and June-August 2021), in addition to the “overall” period of April 2020 - September 2021, is confusing. The scientific rationale for examining the “overall” period in addition to the actual lockdowns is unclear. Additionally, it is not obvious from reading the manuscript whether they actually conducted three separate interrupted time series analyses (comparing baseline to each of these periods), or somehow incorporated all of the time periods into a single analysis.

RESPONSE: Thank you for the comment, first we evaluated the effect of lockdown on HIV outcomes by comparing the pre-lockdown to the overall lockdown period. However, more strict lockdown policies were implemented in response to surges in COVID-19 cases and deaths. We had hypothesized that effects would change as the restrictions were gradually relaxed so we wanted to assess those trends.

2. The segmented regression model correctly includes coefficients for linear time and time since the start of the first lockdown, but I believe that it should also include a non-linear coefficient indicating whether the period is prior to or during lockdown (See Jo Y, et al JIAS 2021, or Ross J, et al JIAS 2019).

RESPONSE: We thank the reviewer for this comment and agree that inclusion of this coefficient is necessary for many interrupted time series analyses where the intervention had a sudden and substantial impact on the outcome (described as a level change in the Ross et al, JIAS 2019 manuscript). However, we did not observe this in our study and thus, decided to exclude this coefficient. Despite not including this coefficient, we believe that our impact estimates of the lockdown would have remained the same, as our estimates of the expected trends were derived from models that used pre-lockdown outcomes only and impact estimates were calculated based on the difference between expected and observed trends. 

3. What was the rationale for stratifying the analysis by the site (Mbarara and Masaka)? Table 1 suggests that the populations were quite similar, and presumably they experienced lockdowns at the same time. So why not combine the data?

RESPONSE: Thank you for the comment. The two sites are supported different implementing partners so we anticipated that the adaptations might be different. Additionally, the viral suppression rates differed by study site, so we decided to present the data separately. 

4. In the Discussion, the authors do a nice job describing steps that the Ugandan HIV program took to counter potential effects of the lockdown on missed visits (e.g. Community Distribution Points, multi-month dispensing), which may have contributed to relatively few missed visits. However, I wonder if the authors also looked at the metric of number of *scheduled* visits in addition to missed visits. It would be useful to know whether fewer visits occurred overall during the lockdowns - if this was the case, but clinical outcomes were not impacted, this would help make a case for lower-barrier DSD models.

RESPONSE: Thank you for raising this concern. The missed visits were computed basing on the appointments (scheduled visits). During the lockdown, scheduling of visits for PWH was not interrupted. Patients who could not make it to the clinic for the visit were advised to seek care from nearby health facilities. However, data for these patients were still captured in the database. We include this in the discussion section page, line, as a possible explanation as to why there was no significant effect of the lockdown on missed visits (page 15, line 257)

5. Similarly, it is promising to see high levels of viral suppression, but it is possible that a substantially fewer number of people had a viral load *measured* during the lockdown periods. It would be helpful for the authors to present data on the number of viral loads in addition to the proportion suppressed.

RESPONSE: Thank you for raising this concern. We have noted this limitation and added it to the discussion section as follows “Third, we examined the effect of the lockdown on viral suppression but did not assess the effect on viral load coverage” (page 18, lines 282-284)

Minor comments:

6. Intro, paragraph 2, line 73: Can the authors briefly note the findings of the cited studies (references 7 and 11)?

RESPONSE: Thank you, we have included the findings from the two studies as follows “In Uganda, studies conducted to evaluate the effect of the lockdown on HIV care showed an increase in missed visits by 15-50% and no effect on viral suppression” (page 5, lines 74-75)

7. The authors should not refer to patients as “participants” in the manuscript, as they utilized routine clinical data with a waiver of informed consent.

RESPONSE: Thank you for the comment. We appreciate your feedback. We have revised the word “participants” to Persons living with HIV (PWH) in all areas of the manuscript

8. In the Methods the authors should describe/define all variables that were extracted from the EMR (e.g. sex, age, marital status) in addition to outcomes.

RESPONSE: Thank you for the comment. We now include all the variables that we extracted from the EMR in the methods section (page 6, lines 109-110).

9. In Table 1, was the mean number of ART clinic visits extracted over the entirety of the study, or some sort of baseline measurement? If this was over the entire study period, as noted above, would be helpful to see the mean number pre-lockdown and then during the lockdown. Similarly, was duration on ART a baseline value?

RESPONSE: Thank you for this comment. We considered the mean number of ART clinic visits and mean ART duration for the entire study period. We considered a cohort of participants who initiated into care from March 2017 to September 2021. We have revised Table.1 to indicate the pre- and post-lockdown values as well and included a footnote for duration of ART describing that it was for the entire study period. 

10. In Table 2, it would be helpful to have a footnote for “Expected” indicating that this value was derived from the linear regression model

RESPONSE: Thank you for your helpful suggestion, we have agreed to the suggestion and added the footnote “*Expected” on Table 2, page 10, line 185

11. The authors state that all data is available but this does not seem to be the case.

RESPONSE: Data is available on request by the reviewer. Due to the participant privacy policy and use of patient data , we are able to provide access link of the data to the reviewer and PLOS ONE team when requested as opposed to making it available to the repository

12. Reviewer #2: Thank you for the opportunity to review this well written manuscript. The study design and analysis approach is appropriate. The ethical considerations were made. The authors have endeavoured to provide contextual information on how the lockdowns were implemented, the study setting and the type of HIV program innovations which are all critical in understanding the findings of this study. The study strengths and limitations are described.

13. The only question that remains is how other facilities, which are not part of the IeDEA collaboration, would perform with regard to implementing innovations for retention for people living with HIV/AIDS.

RESPONSE: Thank you for highlighting this limitation which we have now clarified under the discussion section to include similar adaptations described in none IeDEA low level facilities page 17/line 287-288

---

## [Decision Letter · Decision Letter 1]

5 Apr 2023

PONE-D-22-34356R1Full Title: Effect of the COVID-19 lockdown on the HIV care continuum in Southwestern Uganda: An interrupted     time series analysisPLOS ONE

Dear Dr. Kabami,

Thank you for submitting your manuscript to PLOS ONE. After careful consideration, we feel that it has merit but does not fully meet PLOS ONE’s publication criteria as it currently stands. Therefore, we invite you to submit a revised version of the manuscript that addresses the points raised during the review process.

 Many thanks for your revision. Please respond as completely as possible to reviewer #1 feedback. We look forward to reviewing your next draft and thank you, again, for choosing to submit to PLOS Global Public Health.

We look forward to receiving your revised manuscript.

Kind regards,

Jake Michael Pry, PhD, MPH

Academic Editor

PLOS ONE

Journal Requirements:

Additional Editor Comments:

We thank you for submitting a revision and responding to the previous reviewer feedback. Upon re-review, reviewer #1 has outlined several remaining edits that should be addressed. We hope that you will choose to revise in accordance with this feedback and submit a revision to PLOS Global Public Health.

Reviewers' comments:

Reviewer's Responses to Questions

**Comments to the Author**

1. If the authors have adequately addressed your comments raised in a previous round of review and you feel that this manuscript is now acceptable for publication, you may indicate that here to bypass the “Comments to the Author” section, enter your conflict of interest statement in the “Confidential to Editor” section, and submit your "Accept" recommendation.

Reviewer #1: (No Response)

2. Is the manuscript technically sound, and do the data support the conclusions?

Reviewer #1: Yes

3. Has the statistical analysis been performed appropriately and rigorously? 

Reviewer #1: I Don't Know

4. Have the authors made all data underlying the findings in their manuscript fully available?

Reviewer #1: No

5. Is the manuscript presented in an intelligible fashion and written in standard English?

Reviewer #1: Yes

6. Review Comments to the Author

Reviewer #1: The authors were responsive to a number of critiques in their revision and the manuscript is improved. However, I do not feel that they adequately responded to several major critiques:

1. The authors have clarified that they examined three different policy periods (overall lockdown, first intense lockdown, second intense lockdown) because of the intensity of the restrictions. However, it remains unclear how this was incorporated into the analysis, as the segmented regression model only includes a term for post overall lockdown period, and no term for the more intense/shorter lockdowns.

2. Another statistical concern - I am not sure that the justification for not including a level change coefficient in the segmented regression model is appropriate/sufficient. The manuscript would benefit from review by a statistician to ensure the analyses were performed correctly.

3. The justification for analyzing Mbarara and Masaka separately is somewhat weak and is not described in the manuscript at all. At the very least, the authors could include this justification in the methods when describing how the analyses were performed.

7. PLOS authors have the option to publish the peer review history of their article (what does this mean?). If published, this will include your full peer review and any attached files.

Reviewer #1: No

---

## [Author Response · Author response to Decision Letter 1]

29 Apr 2023

Title: Effect of the COVID-19 lockdown on the HIV care continuum in Southwestern Uganda: An interrupted time series analysis

Manuscript Number: PONE-D-22-34356

Reviewer #1: The authors were responsive to a number of critiques in their revision and the manuscript is improved. However, I do not feel that they adequately responded to several major critiques:

 The authors have clarified that they examined three different policy periods (overall lockdown, first intense lockdown, second intense lockdown) because of the intensity of the restrictions. However, it remains unclear how this was incorporated into the analysis, as the segmented regression model only includes a term for post overall lockdown period, and no term for the more intense/shorter lockdowns.

RESPONSE: We agree with the reviewer and have added level change and slope change terms to our ITS models to allow for a more accurate examination of the different policy periods. The Methods section has been edited to reflect the new model specification (in Page 8, lines 133-144 and provided below) and revisions have been made throughout the manuscript text, Tables, and Figures. 

“The segmented regression model used to estimate pre- and post-trends was specified as follows: Y_t=β_0+β_1 T_t+ β_2 X_t+ β_3 A_t+β_4 M_t+ β_5 V_t+β_6 C_t, where for time points t, Y represents the outcome; T is a linear term indicating the number of months since the start of the study period (models the pre-lockdown trend); X is a binary term indicating the period after the first lockdown (models level of change first lockdown); A is a linear term indicating the number of months since the start of the first lockdown policy in April 2020 and values prior to April 2020 would be equal to zero (models the slope change of the first lockdown); M is a binary term indicating the period after the second lockdown (models level-change of second lockdown); V is a linear term indicating the number of months since the start of the second lockdown in June 2020 and values prior to June 2020 would be equal to zero (models the slope change of the second lockdown); and C is an indicator variable for calendar month (e.g., January, February, March etc.) to account for seasonality.” 

 Another statistical concern - I am not sure that the justification for not including a level change coefficient in the segmented regression model is appropriate/sufficient. The manuscript would benefit from review by a statistician to ensure the analyses were performed correctly.

RESPONSE: Kindly see response to Reviewer #1, Comment #1. 

 The justification for analyzing Mbarara and Masaka separately is somewhat weak and is not described in the manuscript at all. At the very least, the authors could include this justification in the methods when describing how the analyses were performed.

RESPONSE: We thank the reviewer for this comment. We have provided the following justification in the Methods Section (Page 7, lines 122-126) 

“Prior to conducting ITS analyses, we found substantial differences in pre-intervention outcomes (e.g., number of ART initiations, viral suppression rates) between study sites. Thus, ITS analyses were conducted separately for Mbarara and Masaka Referral Hospitals to explore whether effects of the lockdown varied between study sites.”

---

## [Editor Report · Decision Letter 2]

3 May 2023

PONE-D-22-34356R2Full Title: Effect of the COVID-19 lockdown on the HIV care continuum in Southwestern Uganda: An interrupted     time series analysisPLOS ONE

Dear Dr. Kabami,

Thank you for submitting your manuscript to PLOS ONE. After careful consideration, we feel that it has merit but does not fully meet PLOS ONE’s publication criteria as it currently stands. Therefore, we invite you to submit a revised version of the manuscript that addresses the points raised during the review process.

 Please fully address remaining comments regarding methods and analysis from the editor below. Many thanks for your time.

We look forward to receiving your revised manuscript.

Kind regards,

Jake Michael Pry, PhD, MPH

Academic Editor

PLOS ONE

Journal Requirements:

Additional Editor Comments:

We thank you for your second revision however, the response regarding the statistical soundness of the model has not been fully addressed. The authors indicate comparison by lockdown period however, the model does not appear to allow for such comparisons, per the previous reviewer comment. It might be reasonable to modify methods to indicate models across the lockdown periods however, further evidence that the proposed model allows for the comparisons across lockdown periods specified is needed, both here with the editors and to the readers in the manuscript. Many thanks for your time and we look forward to reviewing you next revision.

---

## [Author Response · Author response to Decision Letter 2]

26 Jun 2023

Response to Editor’s Comments

Please find below our response to the Editor’s comment. In our response, we have provided the page and line numbers where changes have been made to the manuscript. 

Comment: We thank you for your second revision however, the response regarding the statistical soundness of the model has not been fully addressed. The authors indicate comparison by lockdown period however, the model does not appear to allow for such comparisons, per the previous reviewer comment. It might be reasonable to modify methods to indicate models across the lockdown periods however, further evidence that the proposed model allows for the comparisons across lockdown periods specified is needed, both here with the editors and to the readers in the manuscript. Many thanks for your time and we look forward to reviewing you next revision.

Response: We thank the editor and the reviewer for their comments regarding the proposed approach. To address these comments, we have revised our methods, updated our manuscript accordingly, and added sensitivity analyses using a different modelling approach. For these analyses, we sought out additional support from Dr. Yea-Hung Chen, who has expertise in epidemiology and biostatistics (particularly in methods used in this study) and has published several studies in high-impact journals utilizing time series analyses to determine the impact of COVID-19 on mortality1-3 and racial/ethnic disparities.4 Given Dr. Chen’s contributions to the analysis and overall writing of the manuscript, we kindly request he be added as a co-author to our study. In the following sections, we describe in detail our revised methods and the changes made to our manuscript. 

In our revised manuscript, we have re-analyzed our data removing the use of segmented regression model. Rather, our study uses a different statistical approach to estimate the impact of the lockdown policies. Our assumption is that the underlying trend in the pre-intervention period would have continued had it not been ‘interrupted’ by the COVID-19 pandemic and its associated lockdown policies.5-7 The statistical model we used in our study utilized pre-intervention data only to establish the underlying pre-intervention trend and used this model to estimate the expected (counterfactual) trend during the post-intervention period. A graphical interpretation is provided below. 

To establish the underlying trend, we modeled pre-intervention monthly data using Poisson regression, as done in previous studies.8,9 We includedThe most parsimonious model we identified was using the following covariates: a continuous value indicating time in months since the start of study observation and fixed terms indicating calendar months to account for seasonality. An alternative model specification was tested and included in the Supplementary Appendix. To estimate the impact of each post-lockdown policy period, observed outcomes during each post-lockdown month were compared to the expected (counterfactual) values predicted from the model and averaged across the post-lockdown period of interest (during the 1st lockdown, during the 2nd lockdown, and the overall 18-month period after the first lockdown). This described further in our revised Methods section (on Pages 7-8; lines 121-142). 

The current analysis differs from our previous segmented regression approach in one major advantageous way. First, our current model only uses pre-intervention data to predict the expected ‘counterfactual’ trend during the post-lockdown period. Thus, the approach does not rely on modelingthe expected values generated from this model do not rely on intervention-era time (or the policy context during that period). Thus, there is no need to incorporate any model covariates that indicate slope or intercept changes related to each lockdown policy (unlike with the segmented model approach). While this new approach does not evaluate these terms (i.e., slope or intercept changes), we argue that this is not strictly of interest. We are, rather, interested in the cumulative causal effect of the intervention.

While there is no single accepted statistical method for evaluating policy interventions,8 we recognize that the segmented regression approach is athe most common and widely recognized approach.5,10,11 Thus, for clarity, we have amended our manuscript throughout to describe our study as a ‘time series’ analysis. Moreover, given the true counterfactual trend is unverifiable and estimated expected trends are dependent on the choice in statistical model, we added text in our Limitations section that estimates from our analyses should be interpreted with caution (Pages 16-17; lines 275-282 and provided below):

“Third, impact estimates are highly dependent on the statistical model used to establish expected trend. We conducted sensitivity analyses using an ARIMA model which showed similar results for most outcomes, except for two outcomes; ARIMA models suggest COVID-19 was associated with a greater reduction in ART initiations in Masaka Referral Hospital and more missed visits in Mbarara Referral Hospital compared to our current approach. While we believe that our estimates of the expected trend are more plausible for reasons outlined in Supplementary Appendix 1, the estimates produced for these outcomes should be interpreted with caution (as the true counterfactual trend is unverifiable).”

References

1. Chen Y-H, Matthay EC, Chen R, et al. Excess Mortality in California by Education During the COVID-19 Pandemic. Am J Prev Med 2022; 63(5): 827-36.

2. Riley AR, Chen Y-H, Matthay EC, et al. Excess mortality among Latino people in California during the COVID-19 pandemic. SSM Popul Health 2021; 15: 100860.

3. Chen Y-H, Glymour MM, Catalano R, et al. Excess mortality in California during the coronavirus disease 2019 pandemic, March to August 2020. JAMA Intern Med 2021; 181(5): 705-7.

4. Chen R, Aschmann HE, Chen Y-H, et al. Racial and ethnic disparities in estimated excess mortality from external causes in the US, March to December 2020. JAMA Intern Med 2022; 182(7): 776-8.

5. Bernal JL, Cummins S, Gasparrini A. Interrupted time series regression for the evaluation of public health interventions: a tutorial. Int J Epidemiol 2017; 46(1): 348-55.

6. Morgan SL, Winship C. Counterfactuals and causal inference: Cambridge University Press; 2015.

7. Kontopantelis E, Doran T, Springate DA, Buchan I, Reeves D. Regression based quasi-experimental approach when randomisation is not an option: interrupted time series analysis. BMJ 2015; 350.

8. Harper S, Bruckner TA. Did the Great Recession increase suicides in the USA? Evidence from an interrupted time-series analysis. Ann Epidemiol 2017; 27(7): 409-14. e6.

9. Woolf SH, Chapman DA, Sabo RT, Weinberger DM, Hill L. Excess deaths from COVID-19 and other causes, March-April 2020. JAMA 2020; 324(5): 510-3.

10. Wagner AK, Soumerai SB, Zhang F, Ross‐Degnan D. Segmented regression analysis of interrupted time series studies in medication use research. J Clin Pharm Ther 2002; 27(4): 299-309.

11. Rossen LM, Branum AM, Ahmad FB, Sutton P, Anderson RN. Excess deaths associated with COVID-19, by age and race and ethnicity—United States, January 26–October 3, 2020. MMWR 2020; 69(42): 1522.

---

## [Decision Letter · Decision Letter 3]

10 Jul 2023

Full Title: Effect of the COVID-19 lockdown on the HIV care continuum in Southwestern Uganda: a time series analysis

PONE-D-22-34356R3

Dear Dr. Kabami,

We’re pleased to inform you that your manuscript has been judged scientifically suitable for publication and will be formally accepted for publication once it meets all outstanding technical requirements.

Kind regards,

Jake Michael Pry, PhD, MPH

Academic Editor

PLOS ONE

Additional Editor Comments (optional):

We would like to thank the authors for the extensive revisions to the analysis.

Reviewers' comments:

Reviewer's Responses to Questions

**Comments to the Author**

1. If the authors have adequately addressed your comments raised in a previous round of review and you feel that this manuscript is now acceptable for publication, you may indicate that here to bypass the “Comments to the Author” section, enter your conflict of interest statement in the “Confidential to Editor” section, and submit your "Accept" recommendation.

Reviewer #1: All comments have been addressed

2. Is the manuscript technically sound, and do the data support the conclusions?

Reviewer #1: (No Response)

3. Has the statistical analysis been performed appropriately and rigorously? 

Reviewer #1: (No Response)

4. Have the authors made all data underlying the findings in their manuscript fully available?

Reviewer #1: (No Response)

5. Is the manuscript presented in an intelligible fashion and written in standard English?

Reviewer #1: (No Response)

6. Review Comments to the Author

Reviewer #1: (No Response)

7. PLOS authors have the option to publish the peer review history of their article (what does this mean?). If published, this will include your full peer review and any attached files.

Reviewer #1: No

---

## [Editor Report · Acceptance letter]

1 Aug 2023

PONE-D-22-34356R3 

Effect of the COVID-19 lockdown on the HIV care continuum in Southwestern Uganda: a time series analysis 

Dear Dr. Kabami:

I'm pleased to inform you that your manuscript has been deemed suitable for publication in PLOS ONE. Congratulations! Your manuscript is now with our production department. 

Kind regards, 

on behalf of

Dr. Jake Michael Pry 

Academic Editor

PLOS ONE